# Periodic Mesoporous Organosilica Nanoparticles for CO_2_ Adsorption at Standard Temperature and Pressure

**DOI:** 10.3390/molecules27134245

**Published:** 2022-06-30

**Authors:** Paul Kirren, Lucile Barka, Saher Rahmani, Nicolas Bondon, Nicolas Donzel, Philippe Trens, Aurélie Bessière, Laurence Raehm, Clarence Charnay, Jean-Olivier Durand

**Affiliations:** 1Capgemini Engineering, 75017 Paris, France; lucile.barka@altran.com (L.B.); saher.rahmani@altran.com (S.R.); 2ICGM, Univ Montpellier, CNRS, ENSCM, 34293 Montpellier, France; nicolas.bondon@umontpellier.fr (N.B.); nicolas.donzel@umontpellier.fr (N.D.); Philippe.Trens@enscm.fr (P.T.); aurelie.bessiere@umontpellier.fr (A.B.); laurence.raehm@umontpellier.fr (L.R.); clarence.charnay@umontpellier.fr (C.C.)

**Keywords:** periodic mesoporous organosilica, sol-gel process, CO_2_ adsorption, carbon capture and storage (CCS) application

## Abstract

(1) Background: Due to human activities, greenhouse gas (GHG) concentrations in the atmosphere are constantly rising, causing the greenhouse effect. Among GHGs, carbon dioxide (CO_2_) is responsible for about two-thirds of the total energy imbalance which is the origin of the increase in the Earth’s temperature. (2) Methods: In this field, we describe the development of periodic mesoporous organosilica nanoparticles (PMO NPs) used to capture and store CO_2_ present in the atmosphere. Several types of PMO NP (bis(triethoxysilyl)ethane (BTEE) as matrix, co-condensed with trialkoxysilylated aminopyridine (py) and trialkoxysilylated bipyridine (Etbipy and iPrbipy)) were synthesized by means of the sol-gel procedure, then characterized with different techniques (DLS, TEM, FTIR, BET). A systematic evaluation of CO_2_ adsorption was carried out at 298 K and 273 K, at low pressure. (3) Results: The best values of CO_2_ adsorption were obtained with 6% bipyridine: 1.045 mmol·g^−1^ at 298 K and 2.26 mmol·g^−1^ at 273 K. (4) Conclusions: The synthetized BTEE/aminopyridine or bipyridine PMO NPs showed significant results and could be promising for carbon capture and storage (CCS) application.

## 1. Introduction

In recent years, a significant increase in the concentration of CO_2_ into the atmosphere, one of the main greenhouse gases, has given rise to serious concerns, and there is a clear need for greater efforts to reduce CO_2_ emissions. Indeed, CO_2_ emissions between 1974 and 2004 were greater than those during the 600,000 previous years [1]. Moreover, the current atmospheric concentration of CO_2_ reached a value of 414 ppm in 2020, the highest value on Earth for 3 million years [2]. At the same time, CO_2_ is an ecological and almost unlimited source of carbon and can be used for many applications such as the production of value-added chemicals and energy products including carbon monoxide [3], methane [4], methanol [5,6], formate [5], methoxide [7], cyclic carbonate or polycarbonate [8], formic acid [9], and urea [10]. Thus, the reduction of CO_2_ in the atmosphere has become a very important research topic.

Initially, liquid amine solutions or aqueous alkanolamines such as ethanolamine (MEA), di-ethanolamine (DEA), di-glycolamine (DGA) and *N*-methyldiethanolamine (MDEA) were widely used in the chemical industry to trap acid gases such as carbon dioxide (CO_2_) from gas mixtures [11]. These solutions are able to react and reversibly form other chemical species such as carbamate, carbonate, bicarbonate, etc. [11]. However, this approach to chemical absorption has many disadvantages, such as huge energy consumption for the absorption and regeneration of amines, the high cost of CO_2_ evaporation, and other problems related to amines including poor stability and oxidative degradation of amines, and interference from impure gases such as SO_2_, NO_2_, and NO [12]. Thus, the use of solid adsorbents has gradually replaced this technique.

Several types of material are currently being studied for their CO_2_ adsorption properties, such as mesoporous silicas, zeolites, activated carbons, amine-based materials, metal oxides, and MOFs (metal–organic frameworks) [13,14,15,16]. The latter have great CO_2_ adsorption capacity but many problems remain, such as long and complex syntheses, poor stability depending on temperature and humidity, and complex separation, as well as prices as high as $10,000 to $15,000 per kilogram [16,17,18,19,20,21].

To overcome the problems of conventional liquid amines, new studies have gradually focused on the synthesis of solid adsorbents modified by amines. Recently, porous silica and periodic mesoporous organosilicas (PMOs) have become popular solid adsorbents for CO_2_ capture [22,23]. An increase in CO_2_ adsorption capacity was observed for PMOs modified with compounds containing amines [24], amines having good affinity with CO_2_, allowing templating diamino-alkyltrimethoxysilanes and leading to lamellar materials after the sol-gel procedure [25]. The impregnation of porous silica adsorbents with amine groups has several advantages such as simple manufacture, low cost, and the ability to load a large number of amines [26]. However, the disadvantage of this technique concerns the amines, which are located at the surface of the porous solids and not inside the pores, which may affect the optimal conditions for CO_2_ adsorption. Other scientists have observed a significant improvement in the CO_2_ adsorption by using mesoporous silica modified with various grafted amines such as 3-(aminopropyl)triethoxysilane and *N*-[3-(trimethoxysilyl)propyl]ethylenediamine [27,28]. Furthermore, the use of amine-functionalized organosilica for CO_2_ capture and valorization is growing significantly [29]. The use of silica-based nanoparticles for the storage of CO_2_ has been less described than the use of bulk materials.

In the course of our work concerning the synthesis and applications of periodic mesoporous organosilica nanoparticles (PMO NPs) [30], we were interested in the preparation, of PMO NPs modified with aminopyridine or bipyridine moieties for CO_2_ capture. We believed nanoparticles could be very efficient due to their very high developed specific surface area. Indeed, a recent publication reported the high potential of bis(triethoxysilyl)ethane-based nanocubes for CO_2_ sequestration [23]. The preparation of bipyridine-based PMOs has been studied by Inagaki, Terasaki, Fontecave and co-workers [31,32,33,34,35,36,37] particularly for CO_2_ photoreduction. However, to our knowledge, pyridine and bypyridine-based PMO nanoparticles have not yet been investigated for CO_2_ capture. Therefore, PMO NPs were prepared through the sol-gel procedure with either 1-(6-aminopyridin-2-yl)-3-(3-(triethoxysilyl)propyl)urea (py), 5,5′-bis(triisopropoxysilyl)-2,2′-bipyridine (iPrbipy) or 5,5′-bis(triethoxysilyl)-2,2′-bipyridine (Etbipy) and 1,2 bis-triethoxysilylethane (BTEE) as the PMO NPs matrix. The synthetized PMO NPs were observed by TEM, chemical functionalization was confirmed by FTIR molecular spectroscopy, and the measurement of CO_2_ adsorption on the PMO NPs was performed in pure CO_2_ gas at 273 K and 298 K.

## 2. Results and Discussion

### 2.1. Effect of the Aminopyridine Moiety on the CO_2_ Adsorption

Three BTEE/aminopyridine PMO NPs were synthetized with 1,2-bis(triethoxysilyl)ethane (BTEE) (Figure 1) and 1-(6-aminopyridin-2-yl)-3-(3-(triethoxysilyl)propyl)urea (Figure 2, silylated aminopyridine, (py), prepared through condensation of 2,6-diaminopyridine with 3-(triethoxysilyl)propylisocyanate) [38]. The three types of nanoparticles were obtained through the sol-gel procedure with co-condensation of both precursors by increasing aminopyridine molar fraction: 6% (pyPMO 6 NPs) and 15% (pyPMO 15 NPs). Reference PMO NPs were synthesized with 100% BTEE (PMO 1 NPs).

FTIR molecular spectroscopy analysis was carried out (Appendix A). The obtained spectra allowed the identification of the BTEE precursor with characteristic absorption bands at 1020 cm^−1^ and 1160 cm^−1^, respectively, corresponding to Si-O-Si and Si-OH bonds. The absorption bands corresponding to the aromatic cycle of the pyridine precursor were also observed around 690–750 cm^−1^ (C=C and C=N bonds), 1600 cm^−1^ (amide II band), and 1690 cm^−1^ (amide I band). The presence of these bands proved the success of the co-condensation between the BTEE and the aminopyridine precursors. Moreover, an increase in the nitrogen content was confirmed by elementary analysis (Appendix A).

PMO 1 NPs showed monodisperse nanoparticles of about 80 nm diameter by TEM (Figure 3a and Table 1) and mesoporosity was clearly visible. Dynamic Light Scattering (DLS) in EtOH (Appendix A) confirmed monodispersity with a hydrodynamic diameter of 200 nm. Compared to reference NPs, the addition of aminopyridine led to more polydisperse systems. The pyPMO 6 NPs showed mean diameters of 120 nm but with a standard deviation of 115 nm by TEM (Figure 4a, Table 1). The nanoparticle hydrodynamic diameters distribution was between 100 and 200 nm but aggregates were observed by DLS (Appendix A) at 900 nm. The pyPMO 15 NPs led to smaller nanoparticles of 100 nm diameter but more polydispersed than PMO 1 NPs, (Figure 5a, Table 1) for which DLS showed polydisperse NPs holding 200–500 nm hydrodynamic diameters with large aggregates in the range of 1–2 µm (Appendix A).

In the case of PMO 1, the shape of the sorption isotherm revealed a type IV isotherm, with a narrow pore size located at 2.7 nm. Its specific surface area was found to be 626 m^2^·g^−1^ (Figure 3b). Higher values of specific surface areas were obtained with pyPMO 6, (918 m^2^·g^−1^, Figure 4b) and interestingly, the pore size was similar, compared to PMO 1 (2.7 nm, Figure 4b). The specific surface area seemed to decrease with the increase in the amino pyridine molar fraction to 15% (around 800 m^2^·g^−1^, Figure 4b and Figure 5b). Note that the contribution of mesopores to the sorption capacity was higher compared to PMO 1, which showed a better structuration of the mesoporosity when using pyridine moieties, even though larger distributed pore sizes from 2.5 to 2.8 nm were observed for the larger aminopyridine fraction materials (Table 1, Figure 4b and Figure 5b).

CO_2_ adsorption measurements were performed at 298 K and 273 K (Figure 6, Table 2). A slight increase in the CO_2_ adsorption capacity with the aminopyridine molar fraction was observed at 298 K. Moreover, an increase of 14% in the CO_2_ adsorption was noticed for the pyPMO 15 (0.92 mmol·g^−1^) having the highest aminopyridine content (15%) compared to the 100% BTEE (0.81 mmol·g^−1^). This shows the potential of adding aminopyridine to BTEE for CO_2_ capture. However, different results were observed at 273 K, where the CO_2_ adsorption for the 100% BTEE was higher than with others PMOs, showing in this case the potential of PMO 1 for CO_2_ capture. CO_2_ adsorption increases with decreasing temperature. Indeed, adsorption being an exothermic process, increasing the temperature of the system decreases the adsorption capacity [39]. In terms of affinity, it is worth looking at low CO_2_ pressure, where the highest interaction sites can be found. The shape of the sorption isotherms is almost linear up to 266 hPa, which confirms the rather poor affinity of these materials for carbon dioxide, regardless of temperature. In these cases, similar Henry’s constants can be derived (~1.5·10^−3^ mmol·g^−1^·Pa^−1^ at 298 K).

### 2.2. Effect of the iPrbipyridine Moiety on the CO_2_ Adsorption

Three BTEE/iPrbipyridine PMO NPs were synthetized with BTEE and 5,5′-bis(triisopropoxysilyl)-2,2′-bipyridine (iPrbipy) (Figure 7) by increasing the iPrbipyridine molar fraction from 6 to 15%: 6% (iPrbipyPMO 6), 10% (iPrbipyPMO 10) and 15% (iPrbipyPMO 15).

FTIR molecular spectroscopy analysis (Appendix A) revealed the success of the co-condensation between the BTEE and the iPrbipyridine precursors, as the pyridine precursors exhibited the same characteristic absorption bands. Indeed, a new absorption band, centered at 750 cm^−1^ appeared. This band can be attributed to the C=N bonds from the two aromatic cycles. An increase in the intensity of this band with the bipyridine molar fraction was observed, due to the four-fold increase in nitrogen content (Appendix A). This offers convincing proof of the success of the polycondensation between the BTEE and the bipyridine precursors.

TEM images (Figure 8a, Figure 9a and Figure 10a) showed highly porous (hollow systems) but polydispersed nanoparticles with a mean diameter of 92 nm (iPrbipyPMO 6) or 111 nm (iPrbipyPMO 10). DLS in EtOH confirmed polydispersity of iPrbipyPMO 6 with two populations centered at 200 nm and 900 nm (nanoparticles and aggregates, respectively, Appendix A). For iPrbipyPMO 10 and iPrbipyPMO 15 the populations were centered between 500–600 nm, which is characteristic of dispersed aggregates of nanoparticles (Appendix A).

The three iPrbiPyPMO materials were mesoporous, as indicated by a clear uptake at intermediate relative pressures followed by saturation (Figure 8b, Figure 9b and Figure 10b, Table 3). However, this saturation became less clear as the bipyridine moieties fraction increased. Concomitantly, the hysteresis loop increased. The shape of the hysteresis loop (H3 type) indicates the presence of hollow materials, in which desorption occurs through small mesopores compared to the size of the cavities. The specific surface area with the addition of 6% and 10% bipyridine was higher compared to that of PMO 1 (958 m^2^·g^−1^ and 796 m^2^·g^−1^). The derivation of the pore size distribution using the desorption branches of the sorption isotherms gave two populations of pores, the most important located at ~2.5 nm and a second one located at 3.7 nm. The latter is reminiscent of cavitation effects when desorption occurs in small mesopores connected to large cavities, which is precisely the case with hollow materials.

In terms of CO_2_ capture, the bipyridine-based PMOs behaved differently compared to the aminopyridine-based PMOs (Figure 11, Table 4). At low pressure, the four traces were different, and they were almost superposed in the case of aminopyridine-based PMOs. In the latter case, the influence of aminopyridine moieties was found to be moderate at low pressure, as the sorption capacity was close to that of PMO 1. Using bipyridine moieties, PMO 1 had the lowest CO_2_ sorption capacity, which indicates a clear interaction between bipyridine moieties and carbon dioxide. Quantitatively, the highest CO_2_ adsorption capacity was obtained with iPrbipyPMO 6 with 6% iPrbipyridine (1.045 mmol·g^−1^ at 298 K and 2.26 mmol·g^−1^ at 273 K). A straightforward explanation is that the greatest specific surface area is found with this material, compared to the other PMOs of the series (Figure 8b, Table 3). In terms of sorption efficiency, it is worth differentiating the sorption capacity per square meter. This calculation yields different sorption capacities: 1.09 µmol·m^−2^, 0.992 µmol·m^−2^, and 1.97 µmol·m^−2^ for iPrbipyPMO 6, iPrbipyPMO 10, and iPrbipyPMO 15, respectively. This trend shows that bipyridine moieties interact with CO_2_, even though other effects may come into play, such as structuration, pore blocking, and so on. At 273 K, it can be noted that the sorption capacities of the bipyridine-based PMOs reach those of PMO 1, which indicates that even if the CO_2_/bipyridine-based PMOs’ affinity is greater compared to PMO 1, their sorption capacities are similar (apart from 6% bipyridine).

### 2.3. Effect of the Etbipyridine Moiety on the CO_2_ Adsorption

Three BTEE/Etbipyridine PMO NPs were synthetized with BTEE and 5,5′-bis(triethoxysilyl)-2,2′-bipyridine (Etbipy) (Figure 12), by increasing the Etbipyridine molar fraction: 6% (EtbipyPMO 6), 10% (EtbipyPMO 10) and 15% (EtbipyPMO 15).

On the IR spectra side, (Appendix A) as for the synthetized PMO NPs with the iPrbipyridine precursor, the characteristic absorption band of the C=N aromatic bond located around 750 cm^−1^ increased with the Etbipyridine molar fraction, a feature of the co-condensation between the BTEE and the Etbipyridine precursors. This band did not appear for the aminopyridine precursor, probably due to the difference in chemical structure with the presence of two aromatic cycles with nitrogen for the iPrbipyridine and the Etbipyridine, against only one aromatic cycle for the aminopyridine precursor. An increase in the percentage of nitrogen with the increase in the molar fraction of Etbipyridine was observed. However, these percentages were lower than with the other precursors (Appendix A).

TEM images (Figure 13a, Figure 14a and Figure 15a) showed the formation of larger particles than pyPMO NPs and iPr PMO NPs, with diameters ranging from 300 to 588 nm. A significant standard deviation was also observed for the three PMO NPs (Table 5). Hollow structure was not observed. DLS analysis revealed a single population of nanoparticles with mean hydrodynamic diameter around 1100 nm, due to aggregation of the NPs in EtOH (Appendix A).

In terms of textural properties (Figure 13b, Figure 14b and Figure 15b), the sorption isotherms obtained with the EtbipyPMO series belong to type IV, as found for the other series studied in this work. In this particular case, the saturation plateaus were very flat, which shows that these materials are purely mesoporous, without side adsorption in voids. This is confirmed by the absence of hysteresis loops for these PMOs, which suggests that they are not aggregated. The mesopore volume decreased from 41% down to 27% for increasing fractions of ethoxybipyridine moieties. At the same time, the surface areas obtained were lower compared to PMO 1 NPs, but also compared to other PMOs studied in this work. This could be due to the greater thickness of the pores’ walls as a consequence of the condensation of the ethoxybipyridine moieties. Additionally, the average mesopore size was found to slightly increase for growing fractions of ethoxybipyridine moieties (from 2.8 nm up to 3.4 nm).

The carbon dioxide sorption isotherms obtained with this series of PMOs strongly depended on the fraction of Etbipyridine fraction (Figure 16, Table 6). This trend was even more pronounced compared to the iPrbipyridine series. Indeed, at low pressure, the slopes were completely different, regardless of temperature, and the uptake at 10^5^ Pa was also different. Intriguingly, iPrbipyridine PMO 10 showed a greater affinity compared to the other PMOs, however this feature was not present at 273 K. Two materials showed a poor CO_2_ capture capacity compared to PMO 1, namely the Etbipyridine 6 and Etbipyridine 10 PMOs. On the other hand, Etbipyridine PMO 15 exhibited better sorption capacities than PMO 1. We further derived the adsorbed amount per square meter to evidence the efficiency of these materials. The adsorbed amounts were found to be 1.2 µmol·m^−2^, 2 µmol·m^−2^, 4 µmol·m^−2^, and 2.5 µmol·m^−2^ for PMO 1 NPs, EtbipyPMO 6 NPs, EtbipyPMO 10 NPs, and EtbipyPMO 15 NPs, respectively, which shows that carbon dioxide interacts with Etbipyridine moieties more than with PMO 1 NPs.

Despite the similarity of the chemical structures between the iPrbipyridine and the Etbipyridine precursors, the CO_2_ adsorption is favored (as for the aminopyridine) for a large Etbipyridine molar fraction (15%), and for a low iPrbipyridine molar fraction (6%).

### 2.4. Selection of the Best PMOs for the CO_2_ Capture

Our results compare well with some of the studies which use amine-modified porous silicas for CO_2_ capture and the field has been comprehensively reviewed [40,41]. Note that these studies concern bulk materials. For instance, in our case, at 1 atm, the adsorbed CO_2_ amount obtained for the best materials is around 1 mmol/g which corresponds to 45 mg/g. Using APTES-grafted MCM-41, Mello et al. obtained 33 mg/g at 20 °C [42] with APTES-grafted MCM-48; Kim et al. obtained 35 mg/g [43]. However, using MCM-41 with larger mesopores grafted with diethanolamine, 104 mg/g of CO_2_ could be adsorbed at very low pressure (0.05 atm) [44].

Under more favourable conditions (lower temperature [45], higher pressure [42], or highly selective polyethyleneimine [46]), larger CO_2_ amounts have been captured (135 mg/g).

Finally, according to the different types of synthesized PMO NPs, the highest CO_2_ adsorption values were obtained at 298 K for:iPrbipyPMO 6 (94% BTEE/6% iPrbipyridine): 1.04 mmol·g^−1^;EtbipyPMO 15 (85% BTEE/15% Etbipyridine): 0.95 mmol·g^−1^;pyPMO 15 (85% BTEE/15% aminopyridine): 0.92 mmol·g^−1^.

Compared to the 100% BTEE (0.81 mmol·g^−1^), higher CO_2_ adsorption capacity was observed with 29% (iPrbipyPMO 6 NPs), 17% (EtbipyPMO 15 NPs) and 14% (pyPMO 15 NPs), which showed the strong impact of the polycondensation between the BTEE and the precursor containing aminopyridine groups, amines having an important affinity with the CO_2_ molecule. On the one hand, the best CO_2_ adsorption values were obtained with the bipyridine and the Etbipyridine precursors, proving the importance of the presence of two aromatic cycles with nitrogen for CO_2_ capture. On the other hand, CO_2_ adsorption was favored with a large aminopyridine (or Etbipyridine) molar fraction (15%), and with a low iPrbipyridine molar fraction (6%). In terms of CO_2_/PMO affinity, these three materials exhibited different features, as can be seen in Figure 17, which presents plots of the isosteric heat of adsorption.

In the case of the pyridine-based material, the enthalpy of adsorption was rather weak, close to the enthalpy of condensation of CO_2_ at 298 K, that is 16.7 kJ·mol^−1^ at room temperature [47]. Furthermore, the shape of the enthalpic curve was rather flat, which indicates that this material is seen as homogeneous by carbon dioxide species. The other materials were more interesting in terms of affinity, as the enthalpy of adsorption was close to −30 kJ·mol^−1^ at low coverage, which indicates a high CO_2_/PMO interaction. However, the high interaction sites did not dominate the sorption processes, as shown by the fact that the enthalpy curves strongly decrease as coverage increases. This is especially true for Etbipyridine-based PMO, whereas in the case of the iPrbipyridine analogue, there is an inflexion in the curve (located at θ = 0.2) which suggests two sorption regimes. At very low coverage, sorption is likely to mostly happen on the iPrbipyridine moieties, while at higher coverage, sorption proceeds on the whole surface of the material. To investigate these observations, we performed two sorption cycles using the procedure described in the experimental section (Figure 18). However, between the two cycles, iPrbipy PMO 6 was subjected to a re-activation stage at 60 °C for 2 h under vacuum. The two sorption isotherms were found to be close to one another, however with ~15% difference at 10^5^ Pa. This difference can therefore be accounted for by the interaction sites of iPrbipy PMO 6. In the case of the other materials, for which less interaction was found, it can be anticipated that consecutive sorption isotherms would lead to closer sorption isotherms.

It is therefore interesting to discuss the properties of these PMOs not only in terms of their sorption capacity, but also in terms of their sorption efficiency, as these two aspects of the sorption process lead to the choice of the appropriate material. Where traces of CO_2_ are concerned, a high affinity is required, whereas if large amounts of CO_2_ must be adsorbed, large CO_2_ sorption capacities should prevail.

## 3. Materials and Methods

Cetyltrimethylammonium bromide (CTAB), sodium hydroxide, 3-(triethoxysilyl)propylisocyanate, 1,2-bis(triethoxysilyl)ethane (BTEE), 2,6-diaminopyridine, ammonium nitrate (NH_4_NO_3_), and potassium bromide, were purchased from Sigma-Aldrich (St. Louis, MO, USA). 5,5′-bis(triisopropoxysilyl)-2,2′-bipyridine was purchased from TCI. 5,5′-bis(triethoxysilyl)-2,2′-bipyridine was purchased from Sikemia. Absolute ethanol was purchased from Fisher Chemicals. 

### 3.1. Synthesis of BTEENPs

For CO_2_ adsorption, PMO NPs were synthesized using the sol-gel procedure. A total of 250 mg of CTAB as the surfactant (6.86 × 10^−4^ mol), 437 µL of NaOH solution (2 mol·L^−1^), and 333 µL of 1,2-bis(triethoxysilyl)ethane (or BTEE, Figure 1) (9 × 10^−4^ mol) were dissolved in 60 mL of deionized water. The mixture was stirred for 2 h at 80 °C/750 rpm. The reaction mixture was cooled to room temperature and centrifuged for 20 min at 20,000 rpm. The CTAB surfactant was removed by 45 min sonication at 40 °C, with a 30 mL solution of ammonium nitrate (NH_4_NO_3_) in EtOH (6 g·L^−1^), followed by one water wash and two EtOH washes (30 mL each). BTEENPs were dried in a vacuum oven at 70 °C.

### 3.2. Synthesis of BTEE-Aminopyridinenps/BTEE-iPrbipyridineNPs/BTEE-EtbipyridineNPs

With regard to the BTEENPs, the BTEE-aminopyridineNPs, the BTEE-iPrbipyridine NPs and the BTEE-Etbipyridine NPs were synthesized by the sol-gel procedure. The chemical formulas of each precursor are presented in Figure 2, Figure 7 and Figure 12. The synthesis was adapted from [48]. For all the PMOs, a total of 1.04 g of CTAB (2.85 × 10^−3^ mol) and 0.472 g of NaOH solution (1.18 × 10^−2^ mol) were dissolved in 31.8 mL of deionized water. The total number of moles for the two precursors (BTEE/aminopyridine or iPrbipyridine or Etbipyridine) was 5.18 × 10^−3^, the molar fraction being different. For example, with iPrPMO 6 (94% ethane/6% bipyridine), a solution of 1.8 mL of BTEE (4.86 × 10^−3^ mol) and 0.181 g of 5,5′-bis(triisopropoxysilyl)-2,2′-bipyridine (3.20 × 10^−4^ mol) was added and the mixture was stirred for 24 h at room temperature, at 750 rpm. The reaction was stirred for 2 h at 95 °C and 750 rpm. The reaction mixture was cooled to room temperature and centrifuged for 20 min at 20,000 rpm. Surfactant was removed by by 45 min sonication at 40 °C, with a 30 mL solution of ammonium nitrate (NH_4_NO_3_) in EtOH (6 g·L^−1^), followed by one diethyl ether wash and two EtOH washes (30 mL each). Finally, the NPs were dried in a vacuum oven at 70 °C. This process was mainly used with the iPrbipyridine and the Etbipyridine precursors (Figure 6 and Figure 10). Next, 2,6-diaminopyridine was silylated with 3-(triethoxysilyl)propylisocyanate to produce the silylated precursor 1-(6-aminopyridin-2-yl)-3-(3-(triethoxysilyl)propyl)urea (py), the chemical structure of which is presented in Figure 2. The protocol established by Cho et al. was applied for the silylation process [38]. Finally, the BTEE/pyPMOs were synthesized by hydrolysis and polycondensation of 1-(6-aminopyridin-2-yl)-3-(3-(triethoxysilyl)propyl)urea and 1,2(-bis(triethoxysilyl)ethane according to the previous conditions. The process synthesis of the PMO NPs is presented in Figure 19.

### 3.3. DLS Analysis

Dynamic light scattering (DLS) is a non-destructive analysis technique for measuring the hydrodynamic diameter of particles suspended in a liquid. Analysis of the scattering of light from the laser of the device by particles was performed with the Cordouan Technologies DL 135.

### 3.4. FTIR Analysis

Fourier transform infrared spectroscopy (FTIR) is a non-destructive analysis technique based on the absorption of electromagnetic radiation between 2.5 and 25 µm (wavelength between 400 and 4000 cm^−1^). Absorptions in this zone correspond to the movement (vibrations, rotations, etc.) of organic functional groups and can be used to deduce the chemical bonds and the structural details of the material. Analyses were performed using the Spectrum Two FT-IR Spectrometer (Perkin Elmer, Waltham, MA, USA).

### 3.5. TEM Analysis

Transmission electron microscopy (TEM) is an analytical technique which enables observation of the local pore arrangement of a material by means of the interactions that occur when an electron beam accelerated by a high voltage travels through the material. For TEM characterization, the nanoparticles of the material were dispersed in EtOH and carefully deposited on a copper grid with porous carbon films. The JEOL 1400 Plus (120 kV) microscope was used to record the image size, the shapes and the pores of the synthetized nanoparticles.

### 3.6. Textural Analysis

Prior to the sorption measurements, the nanoparticles were evacuated under a secondary vacuum at 80 °C for 8 h. Because our materials are hybrid materials, the organic moieties in the materials cannot withstand high activation temperatures. We optimized the activation temperature, and 80 °C was found to be acceptable. The nitrogen sorption isotherms showed that the mesoporosity is accessible to nitrogen, which supports our choice.

The sorption isotherms were obtained at 77 K using a Micromeritics TriStar device. The specific surface area of the various materials was derived using the BET (Brunauer-Emmet-Teller) method, taking 0.162 nm^2^ as cross sectional area for nitrogen. T-plot analysis of the sorption isotherms revealed that the prepared materials were not microporous, which allowed the use of the BET model for deriving the specific surface areas. The pore size distributions were derived using the BJH (Barrett-Joyner-Helenda) approach on the desorption branches of the sorption isotherms, starting from p/p° = 0.95 downwards. The pore geometry was assumed to be cylindrical.

### 3.7. RMN Analysis

Nuclear magnetic resonance (NMR) spectroscopy is an analytical technique for determining the structure of an organic molecule, exploiting the magnetic properties of certain atomic nuclei, such as hydrogen, carbon, or silicon. Solid ^13^C-NMR analysis was carried out on 50 mg of NPs using a VARIAN 300 MHz (Wild Bore) solid spectrometer (3.2 mm MAS probe) device.

### 3.8. Elemental Analysis

Elemental analysis is a qualitative and quantitative technique for revealing, by means of combustion or pyrolysis, the elemental composition of an organic compound and therefore the mass percentage of elements such as carbon, hydrogen, nitrogen, sulfur and oxygen. Elemental analysis was carried out using an Elementar Vario Micro Cube device with a sample of 15 mg.

### 3.9. CO_2_ Adsorption Measurement

The CO_2_ adsorption isotherms were measured using a Micromeritics 3Flex device. Two sorption temperatures were investigated, namely 273 K and 298 K, in order to provide the isosteric heat of adsorption. Prior to the sorption measurements, the materials underwent a degassing stage at 80 °C for 8 h under a secondary vacuum.

## 4. Conclusions

Various syntheses of PMO NPs were successfully achieved for the CO_2_ capture application. A significant impact on the polycondensation between the BTEE and the aminopyridine or bipyridine (iPrbipyridine, Etbipyridine) precursors was observed, compared to the synthesis of PMO 1 NPs with 100% BTEE. Specifically, an increase in the CO_2_ adsorption of 29% (94% BTEE/6% iPrbipyridine; 1.04 mmol·g^−1^), 17% (85% BTEE/15% Etbipyridine; 0.95 mmol·g^−1^), and 14% (85% BTEE/15% aminopyridine; 0.92 mmol·g^−1^) was observed compared to PMO 1 NPs (0.81 mmol·g^−1^). The addition of aminopyridine or bipyridine groups improved the affinity of the PMO with CO_2_. In conclusion, PMO NPs represent an interesting tool for CO_2_ capture applications. Furthermore, the prepared PMO NPs could be useful for carbonate synthesis from epoxides [49] or hydrogenation of CO_2_ to formic acid or methanol [50].

## Figures and Tables

**Figure 1 molecules-27-04245-f001:**
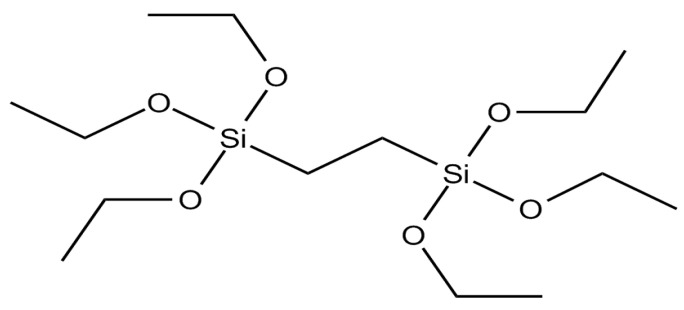
Chemical formula of the 1,2-bis(triethoxysilyl)ethane (BTEE) precursor.

**Figure 2 molecules-27-04245-f002:**
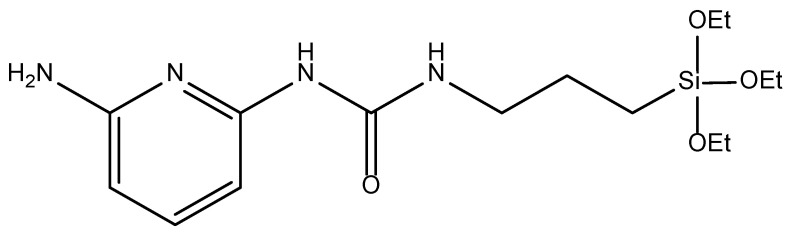
Chemical formula of the 1-(6-aminopyridin-2-yl)-3-(3-(triethoxysilyl)propyl)urea (silylated aminopyridine, py).

**Figure 3 molecules-27-04245-f003:**
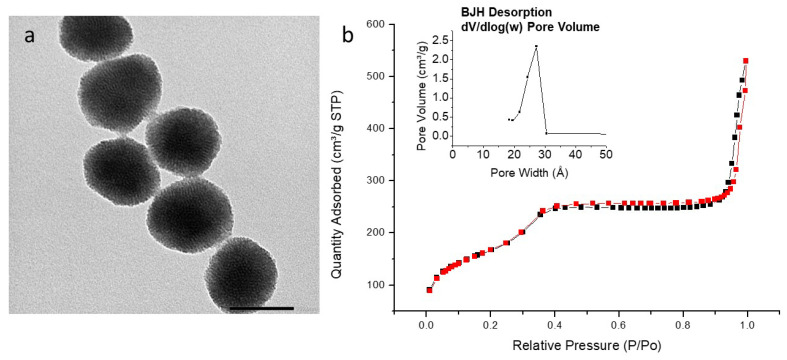
(**a**) TEM image of PMO 1 NPs, scale bar 100 nm. (**b**) Nitrogen adsorption–desorption at 77 K. Insert BJH desorption (dV/dlog(w)) Pore Volume.

**Figure 4 molecules-27-04245-f004:**
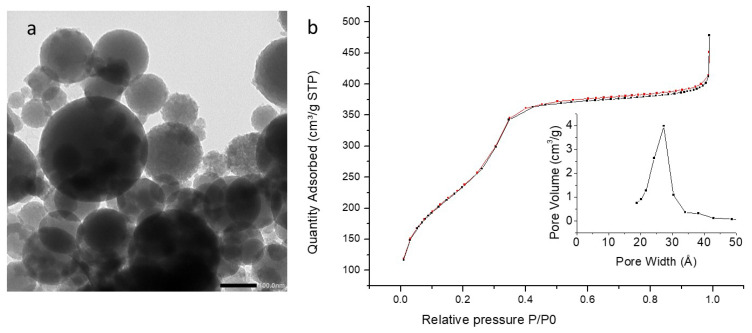
(**a**) TEM image of PyPMO 6 NPs scale bar 100 nm. (**b**) Nitrogen adsorption–desorption at 77 K. Insert BJH desorption (dV/dlog(w)) Pore Volume.

**Figure 5 molecules-27-04245-f005:**
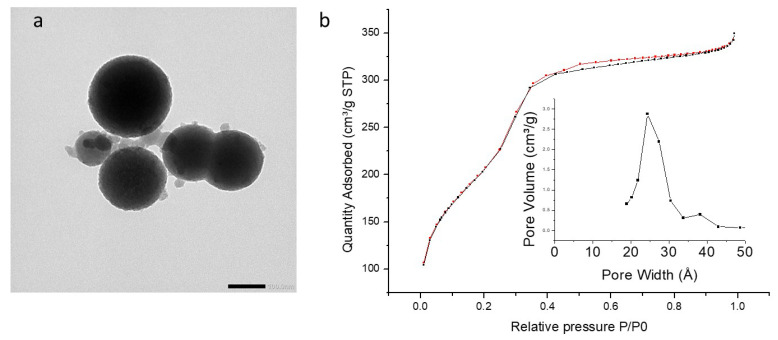
(**a**) TEM image of PyPMO 15 NPs scale bar 100 nm. (**b**) Nitrogen adsorption–desorption at 77 K. Insert BJH desorption (dV/dlog(w)) Pore Volume.

**Figure 6 molecules-27-04245-f006:**
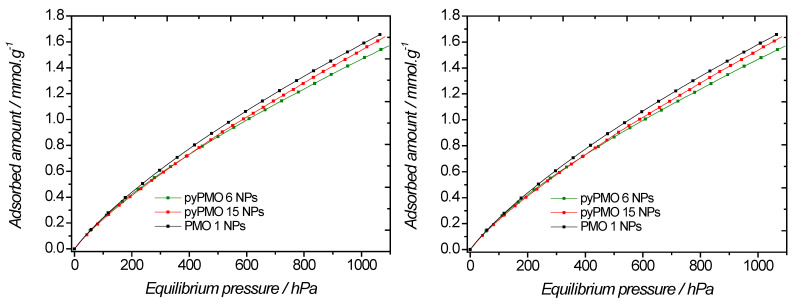
Evolution of the CO_2_ adsorption of BTEE/pyPMO NPs with different pyridine molar fractions versus absolute pressure, at 298 K (**left**) and 273 K (**right**).

**Figure 7 molecules-27-04245-f007:**
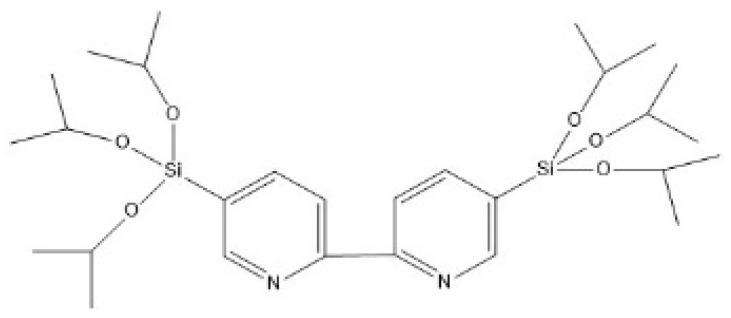
Chemical formula of the 5,5′-bis(triisopropoxysilyl)-2,2′-bipyridine (iPrbipy) precursor.

**Figure 8 molecules-27-04245-f008:**
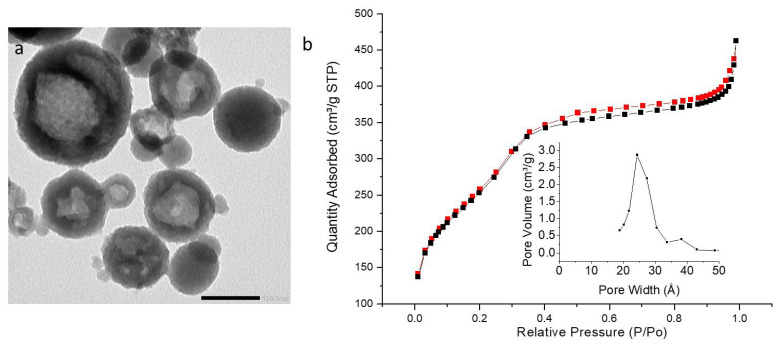
(**a**) TEM image of iPrbiPyPMO 6 scale bar 100 nm. (**b**) Nitrogen adsorption–desorption at 77 K. Insert BJH desorption (dV/dlog(w)) Pore Volume.

**Figure 9 molecules-27-04245-f009:**
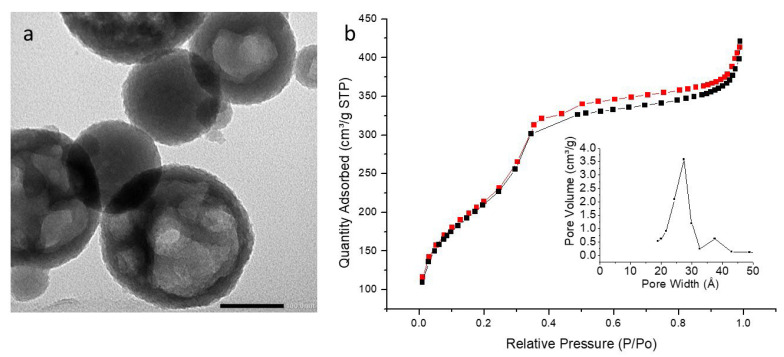
(**a**) TEM image of iPrbiPyPMO 10 scale bar 100 nm. (**b**) Nitrogen adsorption–desorption at 77 K. Insert BJH desorption (dV/dlog(w)) Pore Volume.

**Figure 10 molecules-27-04245-f010:**
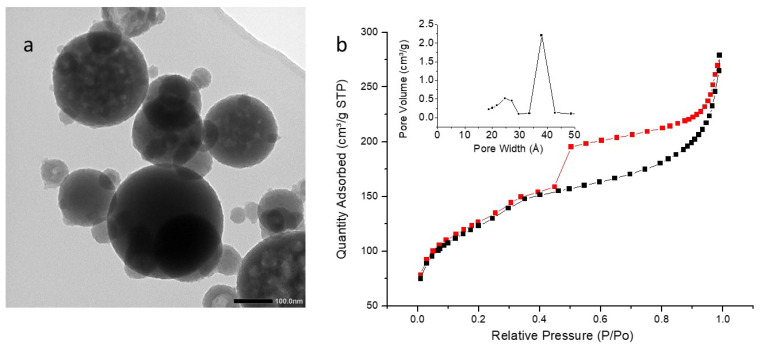
(**a**) TEM image of iPrbiPyPMO 15 scale bar 100 nm. (**b**) Nitrogen adsorption–desorption at 77 K. Insert BJH desorption (dV/dlog(w)) Pore Volume.

**Figure 11 molecules-27-04245-f011:**
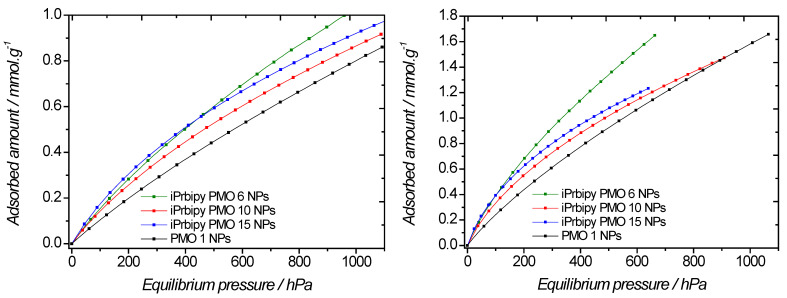
Evolution of the CO_2_ adsorption of BTEE/iPrbipyridine PMO NPs with different bipyridine molar fractions versus absolute pressure at 298 K (**left**) and 273 K (**right**).

**Figure 12 molecules-27-04245-f012:**
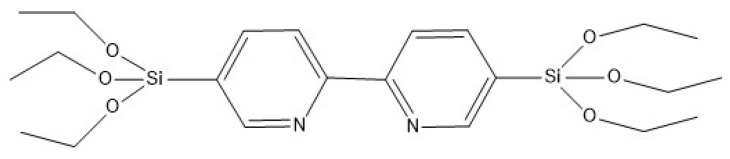
Chemical formula of the 5,5′-bis(triethoxysilyl)-2,2′-bipyridine (Etbipy) precursor.

**Figure 13 molecules-27-04245-f013:**
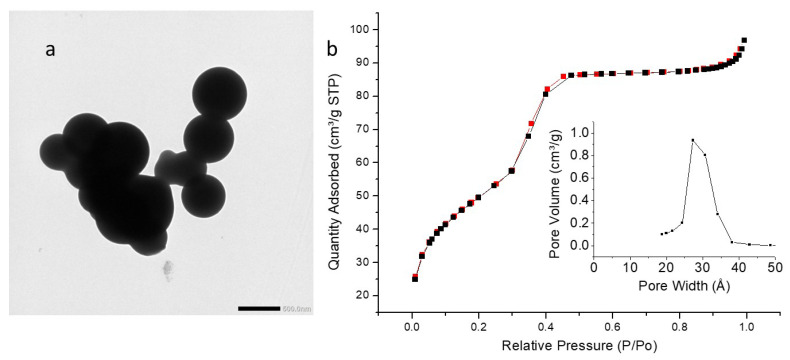
(**a**) TEM image of EtbiPyPMO 6 scale bar 500 nm. (**b**) Nitrogen adsorption–desorption at 77 K. Insert BJH desorption (dV/dlog(w)) Pore Volume.

**Figure 14 molecules-27-04245-f014:**
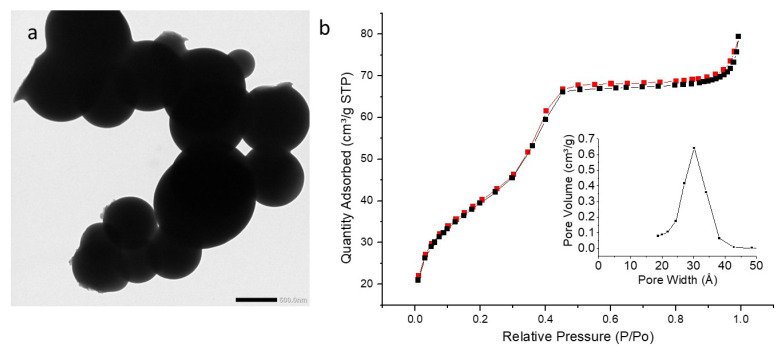
(**a**) TEM image of EtbiPyPMO 10 scale bar 500 nm. (**b**) Nitrogen adsorption–desorption at 77 K. Insert BJH desorption (dV/dlog(w)) Pore Volume.

**Figure 15 molecules-27-04245-f015:**
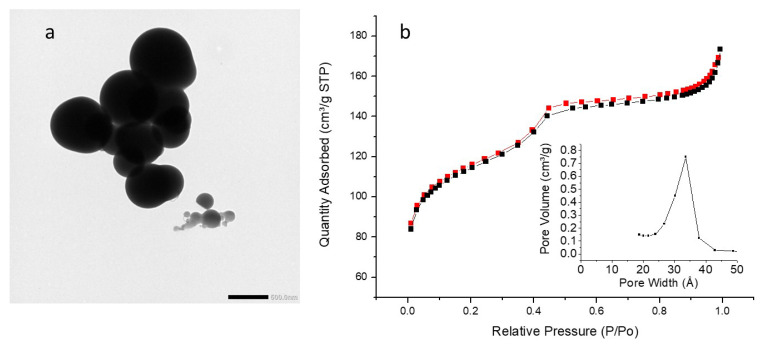
(**a**) TEM image of EtbiPyPMO 15 scale bar 500 nm. (**b**) Nitrogen adsorption–desorption at 77 K. Insert BJH desorption (dV/dlog(w)) Pore Volume.

**Figure 16 molecules-27-04245-f016:**
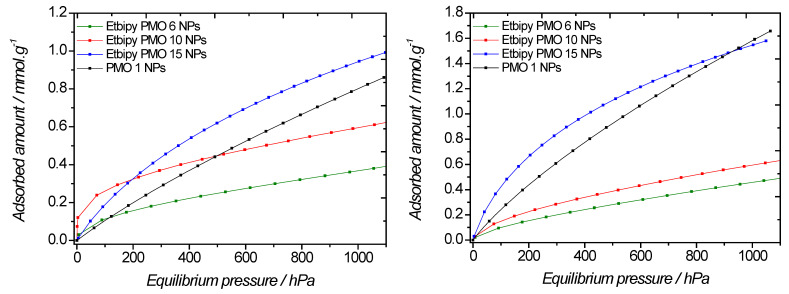
Evolution of the CO_2_ adsorption of BTEE/Etbipyridine PMO NPs with different Etbipyridine molar fractions versus absolute pressure at 298 K (**left**) and 273 K (**right**).

**Figure 17 molecules-27-04245-f017:**
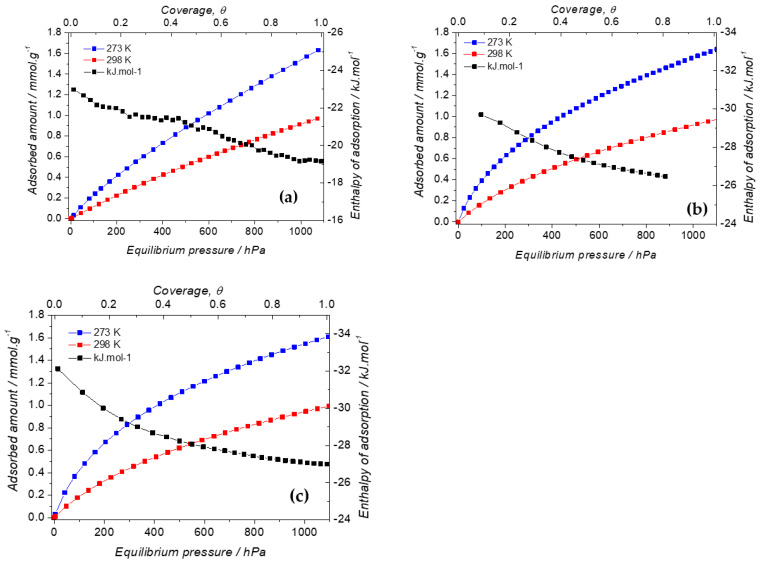
Isosteric heat of adsorption for the selected PMOs: (**a**) pyPMO 15; (**b**) iPrbipy PMO 6; (**c**) EtbipyPMO 15. The maximum coverage, θ, has been defined as the ratio between the adsorbed amount and the adsorbed amount taken at 10^5^ Pa.

**Figure 18 molecules-27-04245-f018:**
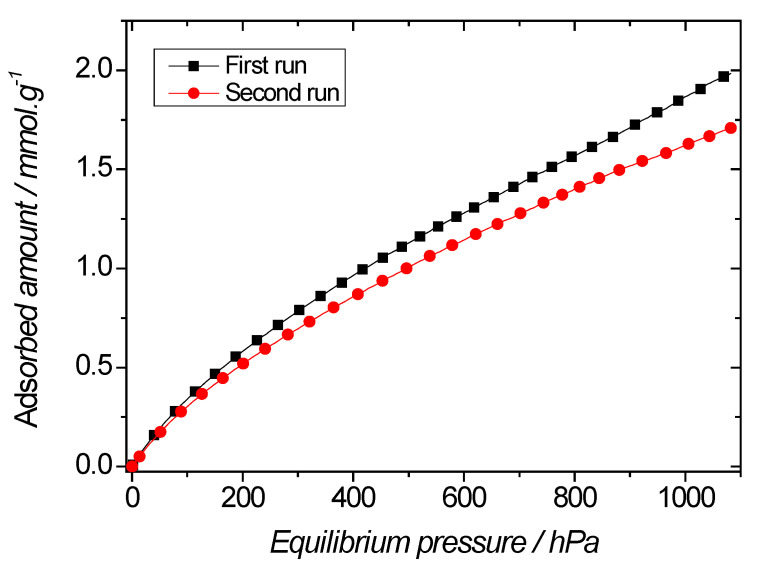
CO_2_ sorption isotherms by iPrbipy PMO 6 at 273 K. (Squares) first sorption isotherm; (Circles) second sorption isotherm after reactivation at 60 °C for 2 h under secondary vacuum.

**Figure 19 molecules-27-04245-f019:**
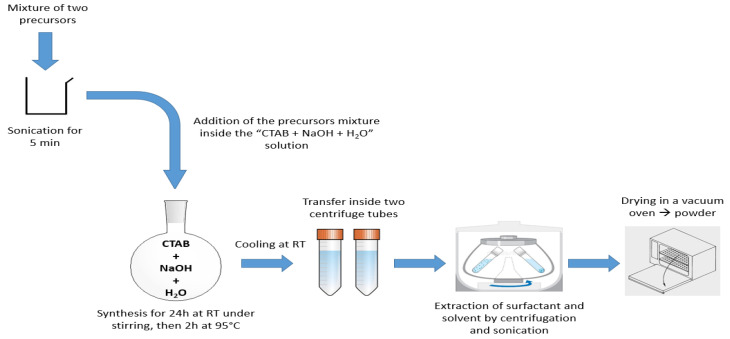
PMO NPs synthesis process with CTAB.

**Table 1 molecules-27-04245-t001:** Size distribution, standard deviation values of PMO 1 NPs, pyPMO NPs, analyzed from TEM, specific surface area and pore sizes of PMO 1 NPs, pyPMO NPs analyzed from N_2_ sorption experiments.

PMO NPs (Composition)	PMO 1 (100% BTEE)	PyPMO 6 (94% BTEE/ 6% Aminopyridine)	pyPMO 15 (85% BTEE/ 15% Aminopyridine)
Particle size distribution (nm)	76	120	99
Standard Deviation (nm)	12	114	45
Specific surface area (m^2^ g^−1^)	626	918	805
Pore size (nm)	2.7	2.7	2.5–2.8

**Table 2 molecules-27-04245-t002:** CO_2_ adsorption capacity (expressed as mmol·g^−1^) at 10^5^ Pa for the BTEE/pyridine PMO NPs.

PMO	T = 298 K	T = 273 K
PMO 1	0.81	1.60
pyPMO 6	0.83	1.48
pyPMO 15	0.92	1.55

**Table 3 molecules-27-04245-t003:** CO_2_ adsorption capacity (expressed as mmol·g^−1^) at 10^5^ Pa for the BTEE/bipyridine PMO NPs.

PMO	T = 298 K	T = 273 K
iPrbipyPMO 6	1.04	2.26
iPrbipyPMO 10	0.79	1.57
iPrbipyPMO 15	0.85	1.56

**Table 4 molecules-27-04245-t004:** Size distribution, standard deviation values of iPrbipyPMO NPs, analyzed from TEM, specific surface area and pore sizes of iPrbipyPMO NPs analyzed from N_2_ sorption experiments.

PMO (Composition)	iPrbipyPMO 6 (94% BTEE/ 6% Bipyridine)	iPrbipyPMO 10 (90% BTEE/ 10% Bipyridine)	iPrbipyPMO 15 (85% BTEE/ 15% Bipyridine)
Particle size distribution (nm)	93	111	102
Standard Deviation (nm)	46	61	72
Specific surface area (m^2^·g^−1^)	958	796	432
Pore size (nm)	2.5	2.5	2.5

**Table 5 molecules-27-04245-t005:** Size distribution and standard deviation values of the BTEE/Etbipyridine PMO NPs analyzed from N_2_ sorption experiments.

PMO (Composition)	EtbipyPMO 6 (94% BTEE/6% Etbipyridine)	EtbipyPMO 10 (90% BTEE/10% Etbipyridine)	EtbipyPMO 15 (85% BTEE/15% Etbipyridine)
Particle size distribution (nm)	301	588	437
Standard Deviation (nm)	287	300	144
Specific surface area (m^2^·g^−1^)	181	143	372
Pore size (nm)	2.8	3.0	3.4

**Table 6 molecules-27-04245-t006:** CO_2_ sorption capacity of the BTEE/Etbipyridine PMO NPs.

PMO	Adsorption at 10^5^ Pa at 298 K (mmol·g^−1^)	Adsorption at 10^5^ Pa at 273 K (mmol·g^−1^)
EtbipyPMO 6	0.37	0.46
EtbipyPMO 10	0.60	0.60
EtbipyPMO 15	0.95	1.66

## Data Availability

Not applicable.

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
