# Peer review of "Periodic Mesoporous Organosilica Nanoparticles for CO2 Adsorption at Standard Temperature and Pressure"

_molecules, 2022, doi:10.3390/molecules27134245_

Round 1

Reviewer 1 Report

The manuscript reports synthesis of mesoporous organosilica nanoparticles and their usage for CO2 adsorption. There are following points need modifications:

1. Please use SI unit.

2. Usually for CO2 capture, CO2 concentration in flue gas is about 10%-30% at atmosphere pressure. That means CO2 adsorped amount in Figure 6 is about 0.1-0.3 mmol/g or in Figure 16 about 0.2-0.5 mmol/g at 293K. Please compare the results with those in available literature, and add discussions. About 2mmol/g is widely reported in literature.

Author Response

The manuscript reports synthesis of mesoporous organosilica nanoparticles and their usage for CO2 adsorption. There are following points need modifications:

  1. Please use SI unit.

We have converted Torr to Hpa

  1. Usually for CO2 capture, CO2 concentration in flue gas is about 10%-30% at atmosphere pressure. That means CO2 adsorped amount in Figure 6 is about 0.1-0.3 mmol/g or in Figure 16 about 0.2-0.5 mmol/g at 293K. Please compare the results with those in available literature, and add discussions. About 2mmol/g is widely reported in literature.

In our study, we decided to focus on low concentration systems under ambient pressure. In a large review, Chen et al gathered all the literature available, regarding CO2 capture by amine functionnalized nanoporous materials. Our results compare well with some of these studies using similar systems, that is amine-modified porous silicas. For instance  in our case, at 1 atm, the adsorbed amount obtained is around 1mmol/g which corresponds to 45 mg/g. Using APTES-grafted MCM-41, Mello et al obtained 37 mg/g  at 20°C (M.R. Mello, D. Phanon, G. Q. Silveira, P. L. Llewellyn and C. M. Ronconi, Micropor. Mesopor. Mater., 143, 174 (2011)) and Kim et al obtained 35 mg/g [S. Kim, J. Ida, V.V. Guliants and J.Y. S. Lin, J. Phys. Chem. B, 109, 6287 (2005)]. However, using MCM-41 with larger mesopores grafted with diethanolamine, 104 mg/g of CO2 could be adsorbed at very low pressure (0.05 atm). R. S. Franchi, P. J. E. Harlick and A. Sayari, Ind. Eng. Chem. Res., 44, 8007 (2005).

Using more favourable conditions (lower temperature [C. Schumacher, J. Gonzalez, M. Perez-Mendoza, P. A. Wright and N. A. Seaton, Ind. Eng. Chem. Res., 45, 5586 (2006)], higher pressure [C. Knofel, J. Descarpentries, A. Benzaouia, V. Zelenak, S. Mornet, P. L. Llewellyn and V. Hornebecq, Micropor. Mesopor. Mater., 99, 79 (2007)], or highly selective ligands [X. Xu, C. Song, J.M. Andresen, B.G. Miller and A.W. Scaroni, Micropor. Mesopor. Mater., 62, 29 (2003)]), larger CO2 amounts have been captured.

We added this comparison in the discussion section of the manuscript (LL272-290).

Reviewer 2 Report

The manuscript "Periodic mesoporous organosilica nanoparticles for CO2 adsorption at standard temperature and pressure" describes the results of ordered micro-mesoporous solids preparation for CO2 adsorption capture.

Main comments:

(i) The conditions of sample preparation before N2 adsorption-desorption analysis is not correct for micro-mesoporous solids analysis.  The temperature of pore fillout has to be 250 C ( doi.org/10.1016/j.colsurfa.2020.125516 ).

(ii) The A(BET) calculations have to revise,  due to micro-mesoporous structure of prepered adsorbents (doi.org/10.1515/pac-2014-1117). 

(iii) How the CO2 equllibrium pressure correlates with provided q(max) and real partial CO2 pressure in air?

(vi) What about desorption study of CO2 and adsorbents stability during multiple adsorption-desorption cycles?

Minor comment:

(i) Please,  avoid the references in Abstract and Conclusions sections. 

(ii) The word "Periodic " should be changed to "Ordered".

Author Response

The manuscript "Periodic mesoporous organosilica nanoparticles for CO2 adsorption at standard temperature and pressure" describes the results of ordered micro-mesoporous solids preparation for CO2 adsorption capture.

Main comments:

(i) The conditions of sample preparation before N2 adsorption-desorption analysis is not correct for micro-mesoporous solids analysis.  The temperature of pore fillout has to be 250 C ( doi.org/10.1016/j.colsurfa.2020.125516 ).

We agree with Reviewer#2 that 250°C is a reasonable temperature for desorbing species from porous materials. This is especially true for microporous materials; such as zeolites or MOFs, in which species such as water can be confined in very narrow micropores. However, our materials are mesoporous, not microporous, which implies that a lower activation temperature can be envisaged. Additionally, our materials are hybrid materials and the organic moieties in the materials cannot stand high temperatures. We optimized the activation temperature, and 80°C found to be acceptable. The nitrogen sorption isotherms show that the mesoporosity is accessible to nitrogen, which supports our choice. We added this comment in the experimental section of the manuscript (LL 365 – 367)

https://doi.org/10.1002/adma.201401931

(ii) The A(BET) calculations have to revise,  due to micro-mesoporous structure of prepered adsorbents (doi.org/10.1515/pac-2014-1117). 

Our materials are not microporous. This could be ensured by t-plot evaluation of the microporosity. The BET model is therefore adequate for measuring the specific surface areas of our materials. We added a comment in the experimental part of the manuscript ( LL 371-373)

 (iii) How the CO2 equllibrium pressure correlates with provided q(max) and real partial CO2 pressure in air?

The CO2 concentration in air is ~400 ppm, which therefore corresponds to the very beginning of our sorption isotherms. This is why we thoroughly described this region of the sorption isotherms for discussing the affinity between CO2 and our materials. In terms of q(max), our materials are therefore able to capture CO2 from high amounts of air, however for such application, PSA is used for recycling the adsorbents when the active sites are saturated by CO2.

(vi) What about desorption study of CO2 and adsorbents stability during multiple adsorption-desorption cycles?

This point is very interesting and we want to thank our reviewer for such suggestion. We performed two adsorption-desorption cycles on a lead material and we demonstrated that this material was able to stand these two cycles without significant loss of efficiency. The difference between the obtained sorption isotherms was accounted for by the presence of strong interaction sites bared by the lead material. A new figure has been added to the manuscript along with the corresponding comments

Reviewer 3 Report

The authors describe the synthesis and characterisation of 3 types of periodic mesoporous organosilica nanoparticles (PMO NPs) and their use for CO2 adsorption.

The manuscript is well planned, and its simple structure/timeliness makes it very easy to read and understand. The discussion is complete and accurately elaborated, describing all the techniques used to characterise the PMO NPs, as well as a plausible application for CO2 adsorption, with only some minor typos throughout the manuscript. The references list is extensive and very complete.

However, the results here presented merit some comments and arise some unresolved questions. 

1. In the abstract I would not add references or titles (Background, methods, results…)

2. It should be explained why BTEE is chosen as the matrix for the nanoparticles

3. The fact that CO2 adsorption of PMO 1 NP at 273 K is the highest and at 298 K is the other way round, should be explained/discussed in more detail.

4. DRX spectra are missed and would give very valuable information regarding crystalline phase, shape anisotropy, strain, and texture (by evaluation of the diffraction peaks’ width, shape, and position) which complements other instrumental characterisation techniques.

Apart from the above, the subject dealt with is very relevant and considerably valuable for the scientific community dedicated to new materials.

Author Response

Please see attachement.

Reviewer 4 Report

In the current study, Kirren et al. design, prepare and characterize three series of organosilica nanoparticles for CO2 adsorption. In each series, they vary the concentration of the monomers (py, Etbipy and iPrbipy) and investigate how this affects the properties and CO2 sorption ability relative to the mother BTEE organopolymer. The study is interesting, the characterization is detailed and carried our consistently. Before accepting the manuscript for publication, the authors should address these points:

1.       In the introduction, the authors do a good job referencing other amine-functionalized systems. They should compare the performance of those systems with their own (for example, a table listing CO2 uptake capacities & conditions).

2.       To confirm the success of condensation, the authors refer to FT-IR results. An explanation should be provided why the C=N bond appears at very different wavenumbers in their three series of materials (690-750, 1600, 600, 500 cm-1). The success of polymerization would be more convincing if the FTIR spectra of the starting materials are added.

3.       How was the particle size distribution determined from TEM images? How many particles were selected and how was unbiased selection ensured? Some numbers appear odd, for instance in Table 1, size 120 nm, standard deviation 114??

4.       Methods section: is the synthesis entirely developed by the authors? If it is based on a published procedure, relevant references should be added.

5.       Some of the figures are incorrectly referred in the main text – please double check.

6.       The DLS data in the Supporting Information should be presented in a better way: the data should be replotted with larger font sizes rather than using screen shots from the DLS software.

Author Response

In the current study, Kirren et al. design, prepare and characterize three series of organosilica nanoparticles for CO2 adsorption. In each series, they vary the concentration of the monomers (py, Etbipy and iPrbipy) and investigate how this affects the properties and CO2 sorption ability relative to the mother BTEE organopolymer. The study is interesting, the characterization is detailed and carried our consistently. Before accepting the manuscript for publication, the authors should address these points:

  1. In the introduction, the authors do a good job referencing other amine-functionalized systems. They should compare the performance of those systems with their own (for example, a table listing CO2 uptake capacities & conditions).

We have added a discussion concerning this point (see reviewer 1).

  1. To confirm the success of condensation, the authors refer to FT-IR results. An explanation should be provided why the C=N bond appears at very different wavenumbers in their three series of materials (690-750, 1600, 600, 500 cm-1). The success of polymerization would be more convincing if the FTIR spectra of the starting materials are added.

We thank the reviewer concerning this point. With aminopyridine moiety, the band at 1690 cm-1 corresponds to amide I signal, the band at 1600 cm-1 corresponds to amide II signal. This has been corrected in the manuscript.

  1. How was the particle size distribution determined from TEM images? How many particles were selected and how was unbiased selection ensured? Some numbers appear odd, for instance in Table 1, size 120 nm, standard deviation 114??

The size of the nanoparticles was measured with Image J softtware. At least 100 nanoparticles were analyzed and in figure 4a the nanoparticles are polydispersed with a mean diameter of 120 nm but the standard deviation is high as some nanoparticles have a size of 50 nm and other nanoparticles have a size of 200 nm.

  1. Methods section: is the synthesis entirely developed by the authors? If it is based on a published procedure, relevant references should be added.

A reference has been added as we adapted a sol-gel procedure.

  1. Some of the figures are incorrectly referred in the main text – please double check.

Indeed, we have corrected this point

  1. The DLS data in the Supporting Information should be presented in a better way: the data should be replotted with larger font sizes rather than using screen shots from the DLS software.

DLS data were revised.

Round 2

Reviewer 2 Report

The revised manuscript could be recommended for the publication.